# The Influence of Malnutrition Measured by Hypalbuminemia and Body Mass Index on the Outcome of Geriatric Patients with a Fracture of the Proximal Femur

**DOI:** 10.3390/medicina58111610

**Published:** 2022-11-07

**Authors:** Bastian Pass, Fahd Malek, Moritz Rommelmann, Rene Aigner, Tom Knauf, Daphne Eschbach, Bjoern Hussmann, Alexander Maslaris, Sven Lendemans, Carsten Schoeneberg

**Affiliations:** 1Department of Orthopedic and Emergency Surgery, Alfried Krupp Hospital, 45276 Essen, Germany; 2Center for Orthopedics and Trauma Surgery, University Hospital Giessen and Marburg, 35043 Marburg, Germany

**Keywords:** hip fracture, albumin, malnutrition, body mass index

## Abstract

*Background and Objectives:* Fractures of the proximal femur are a life-changing and life-threatening event for older people. Concomitant malnutrition has been described as an independent risk factor for complications and mortality. Therefore, we examined the influence of albumin and body mass index (BMI) as parameters for the nutritional state on the outcome after geriatric hip fracture surgery. *Materials and Methods:* Data were retrospectively collected from hospital information systems, and complications and all other parameters were obtained from patient charts. We included patients aged 70 years or above with a fracture of the proximal femur. We excluded periprosthetic and peri-implant fractures and patients with a missing BMI or albumin value. *Results:* Patients with a BMI below 20 kg/m^2^ were more likely to be female but did not differ from the normal BMI group in terms of baseline parameters. Patients with hypoalbuminemia had a higher ASA grade and Charlson Comorbidity Index, as well as a lower hemoglobin value and prothrombin time compared to those with normal albumin values and low BMI. Hypoalbuminemia was associated with significantly increased rates of complications (57.9% vs. 46.7%, *p* = 0.04) and mortality (10.3% vs. 4.1%, *p* = 0.02). Blood loss and transfusion rates were higher in the hypoalbuminemia group. Patients with a BMI below 20 kg/m^2^ had a higher risk of intraoperative cardiac arrest (2.6% vs. 0.4%, *p* = 0.05) but did not show higher mortality rates than patients with a BMI above 20 kg/m^2^. However, the outcome parameter could not be confirmed in the regression analysis. *Conclusions:* Hypoalbuminemia might be an indicator for more vulnerable patients with a compromised hemoglobin value, prothrombin time, and ASA grade. Therefore, it is also associated with higher mortality and postoperative complications. However, hypoalbuminemia was not an independent predictor for mortality or postoperative complications, but low albumin values were associated with a higher CCI and ASA grade than in patients with a BMI below 20 kg/m^2^.

## 1. Introduction

Fractures of the proximal femur are a life-changing and life-threatening event for older people, and despite special treatment protocols, mortality and morbidity are still high [1]. Concomitant malnutrition has been described as an independent risk factor for complications and mortality [2,3,4,5,6]. The prevalence of malnutrition in geriatric trauma patients in Germany is 30–50%, and among such patients, the highest prevalence is seen in patients with a fracture of the pelvis and proximal femur [4,7].

More and more studies are focusing on the influence of malnutrition on hospitalized patients, but only a minority of hospitals are screening for it [4,7,8,9]. Several clinical assessments, laboratory tests, and biometrical parameters are available, and some have been validated for geriatric patients, such as the Nutritional Risk Screening (NRS) and Mini Nutritional Assessment (MNA) [10]. Body mass index (BMI) seems to lack enough sensitivity for standalone screening concerning the prevalence of malnutrition. For example, the obesity paradox describes a positive correlation between survival and body weight in some diseases [11]. Nevertheless, it is nearly always available and easy to handle, so it is often part of malnutrition screening. Recent studies have proposed an optimal geriatric body mass index (BMI) regarding survival, which ranges from 23 to 29.9 kg/m^2^ [12,13].

Another critical point in relation to intervention studies in the field of malnutrition is the validation of outcome parameters. Weight alone, laboratory parameters, arm and ankle diameters, and lymphocyte count do not show a good correlation with nutritional intervention. Despite the knowledge of a potentially modifiable risk factor, surgeons are confronted with the dilemma of the need for prompt surgery, and it is obvious that malnutrition cannot be sufficiently addressed before time-critical surgery. Therefore, we conducted a study to analyze the influence of malnutrition based on BMI and serum albumin on the outcome of patients with a fracture of the proximal femur in a certificated geriatric trauma center.

## 2. Materials and Methods

### 2.1. Data Sources

Data were retrospectively collected from a hospital information system. Complications and all other parameters were obtained from patient charts.

### 2.2. Patients

The inclusion criteria were patients aged 70 years or above with a fracture of the proximal femur from 1 January 2017 to 31 December 2019. A total of 856 patients were included. We excluded periprosthetic and peri-implant fractures and patients with a missing BMI or albumin value. Therefore, each analysis shows the total number of patients that could be included. All included patients were assessed as an emergency. Preoperative fasting was not mandatory. A high-calorie nutrition postoperatively was part of the postoperative care. Albumin or parenteral feeding were not routinely supplied due to low preoperative albumin values. Albumin was not measured postoperatively.

### 2.3. Covariates

The covariates measured included age, gender, BMI, American Society of Anesthesiologist (ASA) Grades 1–5 [14], and the Identification of Seniors At Risk (ISAR) score. The ISAR score ranges from 0 to 6, and the need for geriatric treatment is thought to be necessary for a score of 2 and above [15]. We also examined the use of oral anticoagulants, time to surgery, current residential status, Charlson comorbidity index (CCI), preoperative laboratory results (hemoglobin, leucocytes, thrombocytes, C-Reactive Protein (CRP), International Normalized Ratio (INR), activated Partial Thromboplastin Time (aPTT), creatinine, and albumin), and the presence of injuries besides hip fracture. All complications during the in-hospital stay were measured, and intraoperative processes were analyzed (including central venous catheter, invasive blood pressure measurement, catecholamine, and fluids, as well as blood loss or bone cement implantation syndrome (BCIS)). The perioperative complications were summarized using the Clavien–Dindo classification [16]. Malnutrition was assumed at albumin levels below or equal to 35 g/L or a BMI of 20 kg/m^2^ [17,18].

### 2.4. Outcomes

The primary outcome was mortality during the hospital stay. Other outcome parameters were peri- and postoperative complications measured with the Clavien–Dindo classification, intraoperative fluids, and the need for ICU treatment.

### 2.5. Statistical Analysis

All calculations were performed using the statistics software R v. 4.0.2 (Foundation for Statistical Computing, Vienna, Austria). For descriptive analyses, categorical data were presented as counts and percentages, and continuous variables were presented as the median and interquartile range (IQR). Comparisons between groups were made using the chi² test for categorical variables and the Wilcoxon test for continuous variables. To verify the outcomes of patients according to their albumin value and BMI, patients were sorted into groups.

Linear models and logistic regression models were used to examine the impact of malnutrition on outcomes after controlling for the ISAR grade, ASA grade, sex, age, and type of proximal femur fracture. Results are reported as regression coefficients (ß) for linear regression and odds ratios (OR) for logistic regression, along with the 95% confidence interval (CI). Differences were considered statistically significant at *p* < 0.05.

### 2.6. Ethics

Written patient consent was obtained. This study was performed in accordance with the ethical standards of the 1964 Declaration of Helsinki and its later amendments. According to the guidelines of the responsible state medical association of North Rhine, ethical approval was not necessary for this retrospective anonymous analysis (reference number 70/2021).

## 3. Results

After applying the exclusion criteria, 856 patients remained (Table 1).

Patients with a BMI below 20 kg/m^2^ were more likely to be female and showed significantly decreased albumin and creatinine values than patients with a higher BMI. Hypoalbuminemia was associated with a higher ASA grade, lower BMI, and significantly lower prothrombin time, and patients showed a higher CCI. Significantly lower hemoglobin values were found in both groups. No differences were seen in age, type of fracture, anticoagulants, or ISAR score (Table 2).

### 3.1. Inpatient Outcomes

The inpatient outcome showed no differences in time to surgery or type of surgery in all groups. Hypoalbuminemia was associated with a significantly higher mortality rate and more perioperative complications, as measured with the Clavien–Dindo classification. Neither of the outcome parameters were conspicuous in the BMI analysis. The intraoperative blood loss was inconsistent between the albumin and BMI groups. It was significantly lower in the low-BMI group and in the high-albumin group. The blood transfusion rate was higher in the hypoalbuminemia group. Patients with low albumin values were more likely to receive a central venous line and an invasive blood pressure measurement (Table 3).

### 3.2. Multivariable Logistic and Linear Regression

After controlling for the ISAR grade, ASA grade, sex, age, and type of proximal femur fracture, the impact of BMI and albumin on the outcome parameters showed divergent results. Only a tendency toward higher mortality was seen in patients with hypoalbuminemia (Table 4). An influence of a BMI < 20 kg/m^2^ could not be seen on the outcome parameters (Table 5).

## 4. Discussion

The aim of this study was to examine the differences in outcome between well-nourished and malnourished elderly patients with hip fracture surgery in a specialized orthogeriatric center based on a combination of serum albumin values and BMI. We determined that the nutritional status is associated with the outcome. Patients with low albumin values showed a higher complication rate and mortality rate.

Several observational studies, including a meta-analysis, show that preoperative hypoalbuminemia can predict mortality and morbidity in patients with hip fracture surgery [2,19,20,21,22] and that a poor nutritional state even influences rehabilitation [3]. Our inhouse mortality rate in the hypoalbuminemia group was 10.3%, while it was 4.1% in those with normal values (*p* = 0.02). Comparative studies revealed similar mortality rates of 9.94% vs. 5.53% [19] and 11% vs. 4% [22]. However, it should be taken into account that many other factors, such as sex or age, have an influence on the mortality rate in geriatric hip fractures [23]. We conducted a regression analysis to determine the influence of hypoalbuminemia on the outcome. After controlling for the ISAR grade, ASA grade, sex, age, and type of proximal femur fracture, the effect of albumin on mortality was weaker but still measurable at the significance level of 0.1.

Additionally, it should be mentioned that patients with low serum albumin levels had lower preoperative hemoglobin levels and lower prothrombin time, which is most likely to be seen as an expression of a deficiency-induced synthesis disorder. Both pathologies are reflected in the intra- and postoperative blood loss and transfusion rates, which were significantly higher in the hypoalbuminemia group. As we did not find any differences in the intake of anticoagulants between both groups, the lower prothrombin time could be discussed as an expression of malnutrition as well, as mentioned above, as the parameters measured by the prothrombin time are a product of liver synthesis and are directly dependent on nutrition and vitamin K intake [24].

Data regarding the relationship of a high or low BMI with the outcome parameters after hip fracture surgery are comparatively rare. It has been shown that a BMI below 20 kg/m^2^ is associated with a higher mortality rate compared to normal weight [17,25]. A meta-analysis, including 11 studies, recently demonstrated an inverse relation of body weight with short-term and long-term mortality after hip fracture surgery [26]. Our data did not reveal any significant differences in the mortality rates related to the BMI, but there was a tendency towards a higher mortality rate in the underweight group (BMI < 20 kg/m^2^) group.

It must be mentioned that the BMI was calculated from the anesthesia form and is sensitive to errors as the body weight is difficult to measure before hip fracture surgery, and the declaration of weight and height was made by the patients themselves. Besides the different amounts of crystalloids intraoperatively, no significant difference between the two BMI groups was seen in our analysis, which could be due to the lower number of patients in the lower-BMI group. Our data hinted towards more intraoperative cardiac arrests and higher resuscitation rates in the undernourished BMI group, but this association should be assessed carefully due to the small sample size.

When comparing both identified groups, we found that patients with a low albumin value had a higher CCI and ASA grade, as well as a lower hemoglobin value and prothrombin time. Looking at the outcome parameters, hypoalbuminemia seems to be a more reliable parameter for predicting the likelihood of complications and mortality than a low BMI level. This correlates with previous findings in the literature, where BMI alone has not been recommended for screening malnutrition in geriatric trauma patients. Additional testing should be conducted, such as the MNA short form and the NRS 2002, which were validated in this group of patients and can be performed in a short period of time, even with cognitively impaired patients. For such patients, a helpful and easy-to-handle guideline has been published by the ESPEN society for nutrition in dementia. This guideline can help in making decisions perioperatively, especially in cases of problems with nutritional intake [27]. Lastly, not only malnutrition, but also osteosarcopenia must be kept in mind and detected in such patients as its influence is crucial for the improvement of postoperative outcomes.

However, our analysis demonstrated that there is an association between mortality and hypoalbuminemia following hip fracture surgery. But nourishment cannot be seen as a single parameter and is embedded in a complex of afflictions of old age that might raise the mortality rate in the elderly and is only one of many parameters in geriatric care that should be addressed. These results should encourage all those involved in elderly care to focus on diet and nutritional status. Further studies should focus on intervention in postoperative care to decrease the mortality rate for undernourished patients as there are hints that nutritional support decreases the mortality rate and surgical site infections after hip fracture surgery [28,29].

## 5. Conclusions

Hypoalbuminemia is associated with a higher risk of complications and mortality following hip fracture surgery and is a more reliable parameter to assess the outcome after hip fracture surgery than the BMI. However, albumin is not an independent factor and has to be seen in the context of comorbidities and other risk factors. Nevertheless, this study has some limitations. First of all, it was a retrospective study, and there are known potential biases associated with such as study, e.g., selection bias or confounding factors. Furthermore, this was a single-center study, and internal procedures may limit the universality of our results and conclusions. However, our data are mostly coherent with recent literature. Another limitation concerns the sample size. More reliable statements for rare events require a bigger study population.

## Figures and Tables

**Table 1 medicina-58-01610-t001:** Baseline data. The number of patients with data is given separately for each parameter. SD, standard deviation; ASA grade, American Society of Anesthesiologists; ISAR, Identification of Seniors At Risk; IQR, interquartile range; BMI, body mass index; mg, milligram; g, gram; kg, kilogram; m, meter; dL, deciliter; CCI, Charlson Comorbidity Index.

Patients	*n* = 856
**Age (years)**	
Mean (SD)	85 (6.7)
**Type of fracture**	*n* = 856
Fracture of the femoral neck	447 (52%)
Pertrochanteric fracture	409 (48%)
**ASA grade**	*n* = 856
Median (IQR)	3 (3–3)
1	3 (0.3%)
2	143 (16.7%)
3	625 (73%)
4	85 (10%)
5	0 (0%)
**BMI (kg/m^2^)**	*n* = 665
Median (IQR)	23.4 (23.4–25.9)
<20	117 (17.6%)
20–24.9	323 (48.6%)
25–29.9	174 (26.2%)
≥30	51 (7.7%)
**Hemoglobin (g/dL)**	*n* = 856
Median (IQR)	12.3 (11.18–13.40)
<8	12 (1%)
8–10	94 (11%)
>10	750 (88%)
**Creatinin (mg/dL)**	*n* = 856
Median (IQR)	0.96 (0.76–1.22)
<1.2	626 (73%)
1.2–2	192 (22%)
>2	38 (4%)
**Sex**	*n* = 856
Male	247 (29%)
Female	609 (71%)
**Anticoagulation**	*n* = 856
Yes	386 (45%)
No	470 (87.2%)
**ISAR-Score**	*n* = 855
Median (IQR)	1 (0–3)
0	331 (39%)
1	134 (16%)
2	129 (15%)
3	122 (14%)
4	88 (10%)
5	43 (5%)
6	8 (1%)
**Albumin (g/L)**	*n* = 642
Median (IQR)	39.8 (36.8–42.5)
>35	538 (84%)
30–34.9	68 (11%)
22–29.9	34 (5%)
<22	2 (0%)
**Prothrombin time (%)**	*n* = 856
Median (IQR)	93.5 (78–108)
<30	394 (46%)
30−80	186 (22%)
>80	276 (32%)
**CCI**	*n* = 856
Median (IQR)	2 (1–3)
0	187 (22%)
1–2	375 (44%)
3–4	205 (24%
≥5	89 (10%)

**Table 2 medicina-58-01610-t002:** Comparison of the baseline data between the patients with BMI greater or lesser than 20 kg/m^2^ and albumin greater or lesser than 35 g/L. The number of patients with data is given separately for each parameter. HA, hemiarthroplasty; THA, total hip arthroplasty; ASA grade, American Society of Anesthesiologists; ISAR, Identification of Seniors At Risk; IQR, interquartile range; BMI, body mass index; mg, milligram; g, gram; kg, kilogram; m, meter; L, liter; dl, deciliter; CCI, Charlson Comorbidity Index. * significant differences.

	BMI < 20 kg/m^2^	BMI ≥ 20 kg/m^2^	*p*-Value	Albumin ≤ 35 g/L	Albumin > 35 g/L	*p*-Value
**Patients**	*n* = 117	*n* = 548		*n* = 107	*n* = 535	
**Age (years)**						
Mean (SD)	85	84	*0.41*	85	85	*0.67*
**Sex**	*n* = 117	*n* = 548		*n* = 107	*n* = 535	
Male	20 (17.1%)	170 (31%)	*<0.01 **	31 (29%)	151 (28.2%)	*0.97*
Female	97 (82.9%)	378 (69%)		76 (71%)	384 (71.8%)	
**Type of fracture**	*n* = 117	*n* = 548		*n* = 107	*n* = 535	
Femoral neck	53 (45%)	299 (54.7%)	*0.09*	49 (45.8%)	288 (53.8%)	*0.16*
Pertrochanteric	64 (55%)	249 (45.3%)		58 (54.2%)	247 (46.2%)	
**Type of surgery**						
**Femoral neck**	*n* = 53	*n* = 299		*n* = 49	*n* = 288	
Nail	1 (1.9%)	6 (2%)		0 (0%)	6 (2.1%)	
HA	48 (90.6%)	272 (91%)	*0.32*	45 (91.8%)	264 (91.7%)	*0.26*
THA	1 (1.9%)	12 (4%)		4 (8.2%)	16 (5.6%)	
Others	3 (5.7%)	9 (3%)		0 (0%)	2 (0.7%)	
**Pertrochanteric**	*n* = 64	*n* = 249		*n* = 58	*n* = 247	
Nail	62 (96.9%)	240 (96.4%)		56 (96.6%)	240 (97.2%)	
HA	2 (3.1%)	8 (3.2%)	*0.91*	2 (3.4%)	6 (2.4%)	*0.82*
THA	0 (0%)	1 (0.4%)		0 (0%)	0 (0%)	
Others	0 (0%)	0 (0%)		0 (0%)	1 (0.4%)	
**Anticoagulation**	*n* = 117	*n* = 548		*n* = 107	*n* = 535	
Yes	43 (37%)	249 (45.3%)	*0.11*	42 (39.3%)	233 (43.6%)	*0.48*
No	74 (63%)	299 (54.7)		65 (60.7%)	302 (56.4%)	
**ASA grade**	*n* = 117	*n* = 548		*n* = 107	*n* = 534	
Median (IQR)	3 (3–3)	3 (3–3)	*0.99*	3 (3–3)	3 (3–3)	*<0.01 **
1	0 (0%)	2 (0.4%)		0 (0%)	1 (0.2%)	
2	19 (16%)	97 (17.7%)		10 (9.3%)	92 (17.2%)	
3	89 (76%)	395 (72%)	*0.75*	74 (69.2%)	395 (73.8)	*<0.01 **
4	9 (8%)	54 (9.9%)		23 (21.5%)	47 (8.8%)	
5	0 (0%)	0 (0%)		0 (0%)	0 (0%)	
**ISAR-Score**	*n* = 117	*n* = 547		*n* = 105	*n* = 534	
Median (IQR)	1 (0–2)	1 (0–3)	*0.46*	1 (0–3)	1 (0–3)	*0.40*
0	49 (41.9%)	227 (41.5%)		40 (38.1%)	214 (40.1%)	
1	24 (20.5%)	82 (15%)		14 (13.3%)	80 (15%)	
2	18 (15.4%)	78 (14.3%)		17 (16.2%)	88 (16.5%)	
3	8 (6.8%)	75 (13.7%)	*0.40*	17 (16.2%)	63 (11.8%)	*0.83*
4	12 (10.3%)	49 (9%)		10 (9.5%)	52 (9.7%)	
5	5 (4.3%)	32 (5.9%)		7 (6.7%)	33 (6.2%)	
6	1 (0.9%)	4 (0.7%)		2 (1.9%)	4 (0.7%)	
**Albumin (g/L)**	*n* = 103	*n* = 486				
Median (IQR)	38.6 (33.8–42.1)	40.1 (37.1–42.7)	*0.01 **			
>35	72 (61.5%)	423 (87%)				
30–34.9	15 (12.8%)	45 (9.3%)	*<0.01 **			
22–29.9	16 (13.7%)	16 (3.3%)				
<22	0 (0%)	2 (0.4%)				
**BMI (kg/m^2^)**				*n* = 97	*n* = 492	
Median (IQR)				22.8 (19.5–23.4)	23.5 (21.1–25.9)	*0.02 **
<20				31 (32%)	72 (14.6%)	
20–24.9				38 (39.2%)	249 (50.6%)	*<0.01 **
25–29.9				21 (21.6%)	134 (27.2%)	
≥30				7 (7.2%)	37 (7.5%)	
**Hemoglobin (g/dL)**	*n* = 117	*n* = 548		*n* = 107	*n* = 535	
Median (IQR)	11.7 (10.7–12.8)	12.4 (11.3–13.5)	*<0.01 **	11.2 (10.2–12.5)	12.4 (11.5–13.5)	*<0.01 **
<8	1 (0.9%)	7 (1.3%)		3 (2.8%)	6 (1.1%)	
8–10	17 (14.5%)	56 (10.2%)	*0.38*	24 (22.4%)	48 (9%)	*<0.01 **
>10	99 (84.6%)	485 (88.5%)		80 (74.8%)	481(89.9%)	
**Prothrombine time (%)**	*n* = 117	*n* = 548		*n* = 107	*n* = 535	
Median (IQR)	98 (80–107)	93 (78–107)	*0.44*	86 (74–102)	94 (80–109)	*<0.01 **
<30	62 (53%)	236 (43.1%)		35 (32.7%)	257 (48.0%)	
30−80	22 (18.8%)	131 (23.9%)	*0.14*	40 (37.4%)	103 (19.3%)	*<0.01 **
>80	33 (28.2%)	181 (33%)		32 (29.9%)	175 (32.7%)	
**Creatinin (mg/dL)**	*n* = 117	*n* = 548		*n* = 107	*n* = 535	
Median (IQR)	0.85 (0.72–1.09)	0.97 (0.77–1.25)	*<0.01 **	1 (0.72–1.37)	0.95 (0.77–1.21)	*0.56*
<1.2	96 (82.1%)	392 (71.5%)		67 (62.6%)	398 (74.4%)	
1.2–2	18 (15.4%)	130 (23.7%)	*0.06*	31 (29%)	115 (21.5%)	*0.03 **
>2	3 (2.6%)	26 (4.7%)		9 (8.4%)	22 (4.1%)	
**CCI**	*n* = 117	*n* = 548		*n* = 107	*n* = 535	
Median (IQR)	2 (1–3)	2 (1–3)	*0.72*	2 (1–4)	2 (1–3)	*0.02 **
0	26 (22.2%)	121 (22.1%)		20 (18.7%)	113 (21.1%)	
1–2	58 (49.6%)	241 (44%)	*0.48*	35 (32.7%)	243 (45.4%)	*0.19*
3–4	26 (22.2%)	131 (23.9%)		35 (32.7%)	130 (24.3%)	
>=5	7 (6%)	55 (10%)		17 (15.9%)	49 (9.2%)	

**Table 3 medicina-58-01610-t003:** Comparison of the treatment data between the patients with a BMI greater or less than 20 kg/m^2^ and albumin greater or less than 35 g/L. The number of patients with data is given separately for each parameter. ASA grade, American Society of Anesthesiologists; ISAR, Identification of Seniors At Risk; IQR, interquartile range; BMI, body mass index; h, hour; HA, hemiarthroplasty; THA, total hip arthroplasty; BCIS, bone cement implantation syndrome; g, gram; kg, kilogram; m, meter; L, liter; dL, deciliter; ml, milliliter; CPR, cardiopulmonary resuscitation; RCC, red cell concentrate; IBP, invasive blood pressure measurement; CVC, central venous catheter; ICU, intensive care unit. * Significant differences.

	BMI < 20 kg/m^2^	BMI ≥ 20 kg/m^2^	*p*-Value	Albumin ≤ 35 g/L	Albumin > 35 g/L	*p*-Value
**Time-to-surgery**						
Mean (h)	17.2	14.8	*0.97*	14.9	15.1	*0.96*
**Type of surgery**	*n* = 114	*n* = 540		*n* = 107	*n* = 525	
Nail	63 (55.3%)	246 (45.6%)	*0.10*	56 (52.3%)	249 (47.4%)	*0.32*
HA	50 (43.9%)	280 (51.9%)	*0.12*	47 (44%)	270 (51.4%)	*0.26*
THA	1 (0.9%)	14 (2.6%)	*0.43*	4 (3.7%)	6 (1.14%)	*0.12*
**BCIS**	*n* = 54	*n* = 306		*n* = 51	*n* = 293	
Yes	16 (29.6%)	82 (26.8%)	*0.79*	18 (35.3%)	80 (27.3%)	*0.32*
No	38 (70.4%)	224 (73.2%)		33 (64.7%)	213 (72.7%)	
**Complications**	*n* = 117	*n* = 546		*n* = 107	*n* = 533	
Dindo = 0	60 (51.3%)	296 (54.2%)	*0.64*	45 (42.1%)	284 (53.3%)	*0.04 **
Dindo ≥ 1	57 (48.7%)	250 (45.8%)		62 (57.9%)	249 (46.7%)	
**Deceased**	*n* = 117	*n* = 546		*n* = 107	*n* =533	
Yes	9 (7.7%)	25 (4.6%)	*0.25*	11 (10.3%)	22 (4.1%)	*0.02 **
No	108 (92.3%)	521 (95.4%)		96 (89.7%)	511 (95.9%)	
**Pneumonia**	*n* = 117	*n* = 545		*n* = 107	*n* = 532	
Yes	9 (7.7%)	33 (6%)	*0.65*	7 (6.5%)	33 (6.2%)	*1*
No	108 (92.3%)	512 (94%)		100 (93.5%)	499 (93.8%)	
**Surgical side infection**	*n* = 117	*n* = 546		*n* = 107	*n* = 533	
Yes	3 (2.6%)	4 (0.7%)	*0.21*	0 (0%)	6 (1.1%)	*0.58*
No	114 (97.4%)	542 (99.3%)		107 (100%)	527 (98.9%)	
**Urinary tract infection**	*n* = 117	*n* = 546		*n* = 107	*n* = 533	
Yes	24 (20.5%)	101 (18.5%)	*0.71*	23 (21.5%)	106 (20%)	*0.81*
No	93 (79.5%)	445 (81.5%)		84 (78.5%)	427 (80%)	
**Blood loss**	*n* = 116	*n* = 546		*n* = 106	*n* = 535	
Median (IQR)	100 (0–200)	100 (0–300)	*0.04 **	200 (0–300)	100 (0–300)	*0.02 **
≤500	114 (98.2%)	513 (93.6%)		95 (89.6%)	510 (95.3%)	
501–1000	2 (1.7%)	31 (5.7%)	*0.26*	10 (9.4%)	22 (4.1%)	*0.06*
1001–1500	0	2 (0.4%)		1 (0.9%)	1 (0.2%)	
≥1501	0	2 (0.4%)		0 (0%)	2 (0.4%)	
**Intraoperative crystalloids (mL)**	*n* = 116	*n* = 548		*n* = 106	*n* = 535	
Median (IQR)	1000 (500–1500)	1000 (1000–1500)	*0.06*	1000 (1000–1500)	1000 (800–1500)	*0.19*
≤500	41 (35.3%)	123 (22.4%)		24 (22.6%)	133 (24.9%)	
501–100	41 (35.3%)	254 (46.4%)	*0.03 **	41 (38.7%)	235 (44%)	*0.49*
1001–1500	22 (19%)	117 (21.4%)		27 (25.5%)	115 (21.5%)	
≥1501	12 (10.3%)	54 (9.9%)		14 (13.2%)	52 (9.7%)	
**Intraoperative Colloids (mL)**	*n* = 116	*n* = 548		*n* =106	*n* = 535	
Median (IQR)	0 (0–500)	0 (0–500)	*0.89*	0 (0–500)	0 (0–500)	*0.16*
≤500	115 (99.1%)	530 (96.7%)		102 (96.2%)	522 (97.6%)	
501–1000	1 (0.9%)	16 (2.9%)	*0.36*	3 (2.8%)	12 (2.2%)	*0.41*
1001–1500	0 (0%)	2 (0.4%)		1 (0.9%)	1 (0.2%)	
≥1501	0 (0%)	0 (0%)		0 (0%)	0 (0%)	
**Intraoperative Catecholamine**	*n* = 116	*n* = 548		*n* = 106	*n* =535	
Yes	99 (85.3%)	476 (86.9%)	*0.76*	98 (92.5%)	461 (86.2%)	*0.11*
No	17 (14.7%)	72 (13.1%)		8 (7.5%)	74 (13.8%)	
**Intraoperative CPR**	*n* = 116	*n* = 548		*n* = 106	*n* = 535	
Yes	3 (2.6%)	2 (0.4%)	*0.05*	1 (0.9%)	4 (0.7%)	*1*
No	113 (97.4%)	546 (99.6%)		105 (99.1%)	531 (99.3%)	
**Transfusion (RCC)**	*n* = 117	*n* = 546		*n* = 107	*n* = 533	
Median (IQR)	0 (0–0)	0 (0–0)	*0.93*	0 (0–1)	0 (0–0)	*<0.01 **
0	91 (77.8%)	428 (78.4%)		72 (67.3%)	423 (79.4%)	
1	9 (7.7%)	27 (5%)	*0.41*	9 (8.4%)	27 (5.1%)	*0.05 **
2	13 (11.1%)	56 (10.3%)		16 (15%)	57 (10.7%)	
≥3	4 (3.4%)	35 (6.4%)		10 (9.3%)	26 (4.9%)	
**IBP**	*n* = 116	*n* =548		*n* = 106	*n* = 535	
Yes	45 (38.8%)	229 (41.8%)	*0.62*	62 (58.5%)	220 (41.1%)	*<0.01 **
No	71 (61.2%)	319 (58.2%)		44 (41.5%)	315 (58.9%)	
**CVC**	*n* = 116	*n* =548		*n* = 106	*n* = 535	
Yes	28 (24.1%)	118 (21.5%)	*0.62*	43 (40.7%)	103 (19.3%)	*<0.01 **
No	88 (76%)	430 (88.5%)		63 (59.4%)	432 (80.7%)	
**Postoperative ICU**	*n* = 116	*n* = 548		*n* = 106	*n* = 535	
Yes	12 (10.3%)	42 (7.7%)	*0.44*	13 (12.3%)	43 (8%)	*0.22*
No	104 (89.7%)	506 (92.3%)		93 (87.7%)	492 (92%)	

**Table 4 medicina-58-01610-t004:** Impact of Albumin on different outcome parameters. Results of the logistic and linear regression analysis. All regression models were adjusted for ASA grade, sex, age, CCI, type of fracture. OD, odds ratio; CI, confidence interval; ICU, intensive care unit.

Impact of Albumin on	*n*	OR (95% CI)	*p*-Value
In-hospital mortality	640	1.96 (0.88–4.36)	*0.10*
Complications (Dindo ≥ 1)	640	1.33 (0.85–2.06)	*0.21*
Intraoperative crystalloids	641	1.46 (0.93–2.30)	*0.10*
Intraoperative colloids	641	1.45 (0.92–2.28)	*0.11*
Postoperative ICU treatment	641	1.29 (0.63–2.62)	*0.98*

**Table 5 medicina-58-01610-t005:** Impact of BMI on different outcome parameters. Results of the logistic and linear regression analysis. All regression models were adjusted for ASA grade, sex, age, CCI, type of fracture. OD, odds ratio; CI, confidence interval; ICU, intensive care unit.

Impact of BMI on	*n*	OR (95% CI)	*p*-Value
In-hospital mortality	663	2.01 (0.86–4.68)	*0.11*
Complications (Dindo ≥ 1)	663	1.12 (0.74–1.71)	*0.59*
Intraoperative crystalloids	664	1.00 (0.64–1.58)	*0.98*
Intraoperative colloids	664	1.15 (0.74–1.78)	*0.52*
Postoperative ICU treatment	664	1.58 (0.76–3.29)	*0.21*

## Data Availability

Not applicable.

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
