# Peer review of "The Influence of Malnutrition Measured by Hypalbuminemia and Body Mass Index on the Outcome of Geriatric Patients with a Fracture of the Proximal Femur"

_medicina, 2022, doi:10.3390/medicina58111610_

Round 1

Reviewer 1 Report

This is a highly interesting manuscript focusing on an important medical problem encountered in the surgery of geriatric patients. It is well written. There are minor revisions which should be addressed by the authors.

1. Line 54: modifiable risk factors rather than removable

2. It would be interesting to know how many patients had to be excluded because of incomplete records to account for a potential selection bias

3. The percentage values in table 1 do not add up, for example the albumin levels do add up to 75%. It should be the percentage of the 642 patients with complete albumin levels in the charts.

4. It is very interesting that the regression analysis was not able to prove a significant influence of low albumin values because the supplementation of enteral nutrition has been shown to reduce complications in patients with low albumin levels:

(J Orthop Surg Res. 2019; 14: 292. Published online 2019 Sep 3. doi: 10.1186/s13018-019-1343-2 Supplementation of enteral nutritional powder decreases surgical site infection, prosthetic joint infection, and readmission after hip arthroplasty in geriatric femoral neck fracture with hypoalbuminemia Yaoquan He,1 Jun Xiao,1 Zhanjun Shi,1 Jinwen He,2 and  Tao Li)

The fact that supplementation reduces the complication rate suggest a causative role of low albumin levels in the development of complications. The authors should address this possibility in the discussion briefly.

Author Response

This is a highly interesting manuscript focusing on an important medical problem encountered in the surgery of geriatric patients. It is well written. There are minor revisions which should be addressed by the authors.

  • Thanks a lot for your comment

  1. Line 54: modifiable risk factors rather than removable
  • We substituted “removable” with “modifiable”.

  1. It would be interesting to know how many patients had to be excluded because of incomplete records to account for a potential selection bias
  • We added the data in 2.2. A total of 856 patients were included.
  1. The percentage values in table 1 do not add up, for example the albumin levels do add up to 75%. It should be the percentage of the 642 patients with complete albumin levels in the charts.
  • We rectified the percentage values.
  1. It is very interesting that the regression analysis was not able to prove a significant influence of low albumin values because the supplementation of enteral nutrition has been shown to reduce complications in patients with low albumin levels:

(J Orthop Surg Res. 2019; 14: 292. Published online 2019 Sep 3. doi: 10.1186/s13018-019-1343-2 Supplementation of enteral nutritional powder decreases surgical site infection, prosthetic joint infection, and readmission after hip arthroplasty in geriatric femoral neck fracture with hypoalbuminemia Yaoquan He,1 Jun Xiao,1 Zhanjun Shi,1 Jinwen He,2 and  Tao Li)

The fact that supplementation reduces the complication rate suggest a causative role of low albumin levels in the development of complications. The authors should address this possibility in the discussion briefly.

  • The suggested study shows interesting and important results and we added the conclusions in our discussion and reference list.

Reviewer 2 Report

Review of manuscript ID: medicina-1971560

Title

The Influence of Malnutrition Measured by Hypalbuminemia and Body Mass Index on the Outcome of Geriatric Patients with a Fracture of the Proximal Femur

Critique

In aging society, hip fracture has been increasing in old age population. Also, fracture management regarding nutrition is important because it is sincerely associated with morbidity and mortality. Therefore, understanding the nutrition for albumin have a archival-value in the field of fracture management.

However, there are some conecerns in this study.

1) To reduce the imbalance regarding the covariate, PS matching(Propensity Score matching) analysis should be needed in this study. Furthermore, multivariate logistic regression analysis may be recommended based on PS score.

2) Preoperative nutrition status (NPO time et al..) and postoperative nutrtional care should be described in material and methods section.

3) What are the surgical procedures at each fracture ? / arthroplasty group and internal fixation group may be different because they have a compltely different rehabilitation program (such as weight bearing et al.)

2) What is inclusion and exclusion criteria for the patients with AIS?

4) At the postoperative management of hip fracture surgery, hypoalbuminemia is commonly observed. In this study, hypo-albuminemina is associated with higher risk of complications and mortality. This study seems logical and agrees that. If albumin level is low, we supply it during the acute perioperative period through albumin supplementation, intravenously. Are these measures included in the study ? Or do you think this albumin supplement can reduce complication ?

Author Response

Critique

In aging society, hip fracture has been increasing in old age population. Also, fracture management regarding nutrition is important because it is sincerely associated with morbidity and mortality. Therefore, understanding the nutrition for albumin have a archival-value in the field of fracture management.

However, there are some concerns in this study.

1) To reduce the imbalance regarding the covariate, PS matching(Propensity Score matching) analysis should be needed in this study. Furthermore, multivariate logistic regression analysis may be recommended based on PS score.

  • Thank you very much for your comment. We do totally see your point and we discussed the issue with our statistician. We did some calculations, but we did not gain any other results. Furthermore, we addressed the imbalance in the baseline data in the multivariate logistic and linear regression models. The models were adjusted for ASA grade, sex, age, CCI and type of fracture.

2) Preoperative nutrition status (NPO time et al..) and postoperative nutrtional care should be described in material and methods section.

  • This is a very interesting point. We did not measure the nil-per-os time explicitly. Surgery was declared as an emergency situation and surgery was done as early as possible. We did not wait for preoperative fasting. The preoperative nutrition status (albumin and BMI) were measured at admission and is mentioned in section 2.3. We added it in the 2.2.
  • All patients were offered a high calorie nutrition postoperatively. We added it in 2.2.

3) What are the surgical procedures at each fracture? / arthroplasty group and internal fixation group may be different because they have a compltely different rehabilitation program (such as weight bearing et al.)

  • We added the data in table 2. All patients were allowed to fully weight bear; therefore the rehabilitation program was the same.

2) What is inclusion and exclusion criteria for the patients with AIS?

  • We described the inclusion and exclusion criteria in section 2.2. No other inclusion or exclusion criteria were defined.

4) At the postoperative management of hip fracture surgery, hypoalbuminemia is commonly observed. In this study, hypo-albuminemina is associated with higher risk of complications and mortality. This study seems logical and agrees that. If albumin level is low, we supply it during the acute perioperative period through albumin supplementation, intravenously. Are these measures included in the study ? Or do you think this albumin supplement can reduce complication ?

  • We do not supply albumin due to low serum albumin values routinely without any other medical conditions that require albumin. I re-checked our data and I did not find a case where albumin was given. Due to its high side effects and the need for strict fluid management and monitoring we would not supply it on ward, beside ICU. We do not have any experience with albumin supply intraoperatively, but we will discuss this topic with the anesthetist.

Reviewer 3 Report

Dear authors,

I was pleased to review the paper entitled " The Influence of Malnutrition Measured by Hypalbuminemia and Body Mass Index on the Outcome of Geriatric Patients with a Fracture of the Proximal Femur"

I think that the article is Well written, and concise.

Author Response

Dear authors,

I was pleased to review the paper entitled " The Influence of Malnutrition Measured by Hypalbuminemia and Body Mass Index on the Outcome of Geriatric Patients with a Fracture of the Proximal Femur"

I think that the article is Well written, and concise.

  • Thanks a lot for your comment

Round 2

Reviewer 2 Report

I think my previous opinion was reflected well. I'm very grateful for this.